# MAGE: Model-Level Graph Neural Networks Explanations via Motif-based Graph Generation

**Zhaoning Yu**
Department of Computer Science
Iowa State University
Ames, IA 50010, USA
znyu@iastate.edu

**Hongyang Gao**
Department of Computer Science
Iowa State University
Ames, IA 50010, USA
hygao@iastate.edu

## Abstract

Graph Neural Networks (GNNs) have shown remarkable success in molecular tasks, yet their interpretability remains challenging. Traditional model-level explanation methods like XGNN and GNNInterpreter often fail to identify valid substructures like rings, leading to questionable interpretability. This limitation stems from XGNN's atom-by-atom approach and GNNInterpreter's reliance on average graph embeddings, which overlook the essential structural elements crucial for molecules. To address these gaps, we introduce an innovative **M**otif-b**A**sed **G**NN **E**xplainer (MAGE) that uses motifs as fundamental units for generating explanations. Our approach begins with extracting potential motifs through a motif decomposition technique. Then, we utilize an attention-based learning method to identify class-specific motifs. Finally, we employ a motif-based graph generator for each class to create molecular graph explanations based on these class-specific motifs. This novel method not only incorporates critical substructures into the explanations but also guarantees their validity, yielding results that are human-understandable. Our proposed method's effectiveness is demonstrated through quantitative and qualitative assessments conducted on six real-world molecular datasets. The implementation of our method can be found at https://github.com/ZhaoningYu1996/MAGE.

## 1 Introduction

Graph Neural Networks (GNNs) (Kipf & Welling, 2016; Xu et al., 2018; Gao et al., 2018; Rong et al., 2020; Sun et al., 2022; Wang et al., 2023) have become increasingly popular tools for modeling data in the molecule field. As an effective method for learning representations from molecule data, GNNs have attained state-of-the-art results in tasks like molecular representation learning (Gao & Ji, 2019; Yu & Gao, 2022a; Fang et al., 2022; Zang et al., 2023), and molecule generation (Bongini et al., 2021; Lai et al., 2021). Despite their growing popularity, questions arise about the trustworthiness and decision-making processes of GNNs.

Explaining GNNs has become a major area of interest in recent years and existing methods can be broadly categorized into instance-level explanations and model-level explanations. Instance-level explanations (Ying et al., 2019; Baldassarre & Azizpour, 2019; Pope et al., 2019; Huang et al., 2020; Vu & Thai, 2020; Schnake et al., 2020; Luo et al., 2020; Funke et al., 2020; Zhang et al., 2021a; Shan et al., 2021; Wang et al., 2021; Yuan et al., 2021; Wang et al., 2022; Yu & Gao, 2022b) aim to pinpoint specific nodes, edges, or subgraphs crucial for a GNN model's predictions on one data instance. Model-level explanations (Yuan et al., 2020; Wang & Shen, 2022) seek to demystify the overall behavior of the GNN model by identifying patterns that generally lead to certain predictions. While instance-level methods offer detailed insights, they require extensive analysis across numerous examples to be reliable. In addition, for instance-level explainer, there is no requirement for an explanation to be valid (e.g., a chemically valid molecular graph). Conversely, model-level methods provide more high-level and broader insights explanations that require less intensive human oversight.

Model-level explanation methods have two main categories: concept-based methods and generation-based methods. Concept-based explanation methods focus on identifying higher-level concepts that significantly influence the model's predictions (Azzolin et al., 2022; Xuanyuan et al., 2023). Also,

these methods establish rules to illustrate how these concepts are interconnected, like using logical formulas (Azzolin et al., 2022). Generation-based models try to learn a generative model that can generate synthetic graphs that are optimized to maximize the behavior of the target GNN (Yuan et al., 2020; Wang & Shen, 2022). Instead of defining the relationships between important concepts with formulas, generation-based models can generate novel graph structures that are optimized for specific properties. This capability is particularly crucial in the field of molecular learning.

Despite the notable benefits of generation-based model-level explanation methods for GNNs, they have received relatively low attention. Presently, the available generation-based model-level explainers for GNNs are somewhat limited in their application, especially in molecular graphs. For example, XGNN (Yuan et al., 2020) explains models by building an explanation atom-by-atom and sets a maximum degree for each atom to maintain the explanation's validity. On the other hand, GNNInterpreter (Wang & Shen, 2022) adopts a more flexible approach by learning a generative explanation graph distribution. However, it ensures the validity of explanations by maximizing the similarity between the explanation graph embedding and the average embedding of all graphs, which is not particularly effective

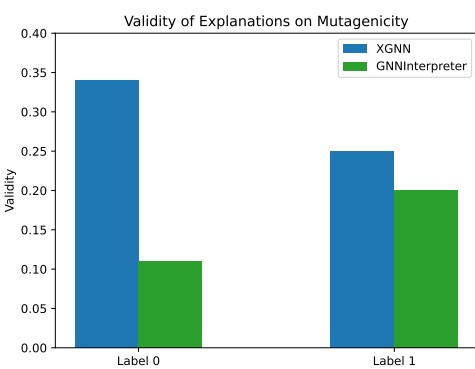

Figure 1: Validity of explanations generated by XGNN and GNNInterpreter on Mutagenicity dataset

for molecular graphs. These existing methods often fail to consider the unique aspects of molecular structures. This oversight makes it challenging for these methods to include crucial substructures in their explanation graphs. It compromises the validity and reliability of the model-level explanations they provide for molecular graphs. In Figure 1, the explanations produced by both XGNN and GNNInterpreter exhibit low validity, making them ineffective for further analysis. Consequently, there's a clear need for more advanced and specialized model-level explanation methods that can effectively handle the unique characteristics of molecular graphs and provide meaningful insights into how GNNs interpret and process molecular data.

This paper introduces a motif-based approach for model-level explanations, MAGE, specifically tailored for GNNs in molecular representation learning tasks. The method begins by identifying all possible motifs from the molecular data. Following this, our method focuses on pinpointing particularly significant motifs for a specific class. This is achieved through a novel attention-based motif learning process, which aids in selecting these key motifs. Once these important motifs are identified, we employ a motif-based graph generator trained to produce explanations based on these motifs. Our experimental results show that our method reliably produces explanations featuring complex molecular structures, achieving complete validity across various molecular datasets. Furthermore, both quantitative and qualitative evaluations demonstrate that the explanation molecules created by our method are more representative and human-understandable than those generated by baseline methods.

The main contribution of our paper can be summarized as below:

1. We propose a novel framework for model-level explanations of GNNs that produces chemical valid explanations on molecular graphs.
2. We develop an attention-based mechanism to extract class-specific motifs, allowing MAGE to provide tailored model-level explanations aligned with the predictive behavior of the target GNNs for each class.
3. We employ motifs as foundational building blocks for model-level explanations generation, offering interpretable and structurally coherent insights into GNN behavior.
4. We demonstrate MAGE's superiority over existing methods through quantitative and qualitative evaluations on multiple molecular datasets.

## 2 PRELIMINARY

**Notations.** A molecular dataset can be depicted as $\mathcal{G} = \{G_1, G_2, ..., G_i, ..., G_n\}$, where each molecular graph within this set is denoted by $G_i = (V, E)$. The label of each graph will be assigned

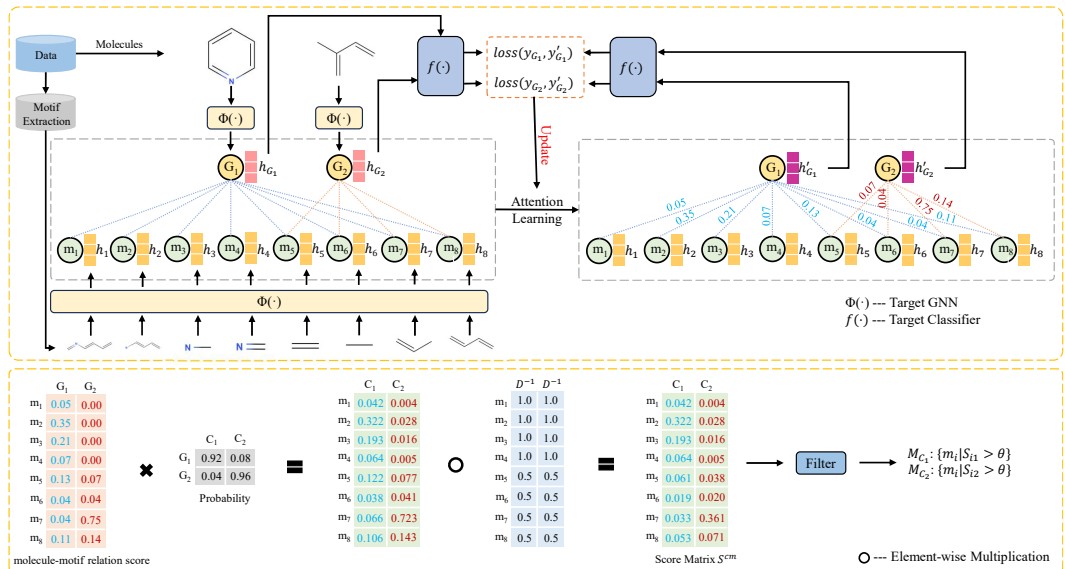

Figure 2: An illustration of the proposed MAGE framework. Given a dataset, a motif extraction algorithm is initially employed to identify all potential motifs. Each motif's feature encoding is derived from the output encoding produced by the target model, which uses the motif graph as its input. A single-layer attention operator is employed to learn the optimal motif combination, maximizing the likelihood that the target classifier will classify both the reconstructed and original molecular encodings identically. To get a score matrix, the method performs a dot product between the attention coefficient matrix and the prediction probability matrix. This score matrix is then normalized using a degree matrix. Finally, motifs whose corresponding scores exceed a specific threshold are selected.

by $l_i \in L = \{l_1, l_2, ...\}$. In this representation, $V$ constitutes a set of atoms, each treated as a node, while $E$ comprises the bonds, represented as edges connecting these nodes. The structural information of each graph $G_i$ is encoded in several matrices. The adjacency matrix $\boldsymbol{A} \in \{0, 1\}^{N \times N}$ represents the connections between atoms, where a '1' indicates the presence of a bond between a pair of atoms, a '0' signifies no bond, and $N$ represents the total number of atoms in the graph. Additionally, an atom feature matrix $\boldsymbol{X} \in \mathbb{R}^{N \times D}$ captures the features of each atom, with $D$ being the number of features characterizing each atom. Similarly, a bond feature matrix $\boldsymbol{Z} \in \mathbb{R}^{|E| \times D_b}$ is used to describe the attributes of the bonds, where $|E|$ represents the number of edges and $D_b$ denotes the number of features describing each bond.

**Graph Neural Networks.** Graph Neural Networks (GNNs) leverage adjacency matrix, node feature matrix, and edge feature matrix to learn node features. Despite the existence of various GNN variants like Graph Convolutional Networks (GCNs), Graph Attention Networks (GATs), and Graph Isomorphism Networks (GINs), they generally follow a message-passing mechanism. In this process, each GNN layer aggregates information from a node's neighbors to refine its representation. For a given hidden layer $i$, the message passing operation can be formulated as $h_i = f(h_{i-1}, \boldsymbol{A}, \boldsymbol{Z})$, where $h_0 = X$. Here, $h_i \in \mathbb{R}^{N \times D}$ is the node representation output from layer $i$, and $D$ is the dimension of the output features. The function $f$ involves computing messages based on the previous layer's node embeddings ($h_{i-1}$), aggregating these messages from neighboring nodes, and updating the hidden representation $h_i$ for each node using the aggregated messages.

**Motif.** A motif is essentially a recurring subgraph within a graph, closely associated with the graph's property, which has been thoroughly researched across various domains, including biochemistry, ecology, neurobiology, and engineering (Milo et al., 2002; Shen-Orr et al., 2002; Alon, 2007; 2019) and has been demonstrated to be of significant importance. GSN (Bouritsas et al., 2022) introduces motif information into node features to improve the learning of GNNs. HMGNN (Yu & Gao, 2022a) generates a heterogeneous motif graph to enhance the molecular representation learning. MotifExplainer (Yu & Gao, 2022b) applies motifs to learn an instance-level explanation for GNNs.

## 3 Motif-based Model-Level GNN Explainer

This work introduces a Motif-bAsed GNN Explainer (MAGE), a novel approach for model-level GNN explanations. This method uses motifs as fundamental elements for interpreting the overarching behavior of a trained GNN for molecular graph classification at the model level. Instead of generating a graph at the atomic level, using motifs allows the generator to assemble graphs from these predefined blocks. This method prevents the creation of potentially invalid intermediate substructures and speeds up the generation process. It achieves this by reducing the number of configurations the generator must consider, focusing solely on the arrangement of motifs rather than the specific placement of each atom and bond.

Given a dataset, denoted as $\mathcal{G}$ with $|\mathcal{G}|$ molecules and $C$ classes, our objective is to generate model-level explanations for a target GNN, which comprises a feature extractor $\phi(\cdot)$ and a classifier $f(\cdot)$. $\phi(\cdot)$ takes a molecular graph or subgraph as input and outputs its feature representation. $f(\cdot)$ utilizes this representation to predict the classification of a molecule.

MAGE begins with identifying all potential motifs within $\mathcal{M}$. Following this, an attention-driven motif learning phase is employed to identify the most significant motifs for each class using $\phi(\cdot)$ and $f(\cdot)$. Finally, a graph generator uses the identified motifs to explain each class.

### 3.1 Motif Extraction

There are mainly four methods for motif extraction. It's important to note that these motif extraction methods can be integrated into our MAGE explanation method, demonstrating its versatility.

1) Ring and bonded pairs (R&B). Methods (Jin et al., 2018; Yu & Gao, 2022a; Bouritsas et al., 2022) in this category recognize simple rings and bonded pairs as motifs. 2) Molecule decomposition. In this category, methods recognize motifs through molecule breakdown. These methods initially establish a catalog of target bonds, either through domain-specific knowledge, such as the "breakable bonds" used by RECAP (Lewell et al., 1998; Zhang et al., 2021b) and BRICS (Degen et al., 2008; Zang et al., 2023; Jiang et al., 2023), or through hand-crafted rules, like "bridge bonds" (Jin et al., 2020). The algorithms remove these target bonds from the molecules and use the consequent molecular fragments as motifs. 3) Molecule tokenization. Certain methods (Fang et al., 2023; Kuenneth & Ramprasad, 2023) leverage the string representations of molecules, such as SMILES, and directly apply the WordPiece algorithm from the NLP domain. The resulting strings within the constructed vocabulary are regarded as motifs. 4) Data-driven method. MotifPiece (Yu & Gao, 2023) statistically identifies underlying structural or functional patterns specific to a given molecular data.

### 3.2 Class-wise Motif Identification

With extracted significant motifs from the previous section, this section introduces an attention-based motif identification approach for each classification class.

**Stage 1: molecule-motif relation score calculation.**
Given the absence of a direct linkage between classification categories and motifs, we propose bridging them through molecules. To this end, we employ an attention operator to learn motif-molecule relation scores using the trained feature extractor $\phi(\cdot)$ and a classifier $f(\cdot)$.

For a given molecule graph $G_i$ and its associated motif set $M_i$, we use an attention operator to identify a combination of motifs that can accurately reconstruct the representations of the molecule graph. The learned attention scores between $G_i$ and motifs in $M_i$ are interpreted as their relation scores. The initial step involves obtaining feature encoding for the molecule and motif graphs. This is done by feeding a molecule graph or motif graph into $\phi(\cdot)$, and the output encoding serves as the molecule or motif:

$$G, \boldsymbol{h}_{m_i} = \phi(m_i), \text{ for all } m_i \in M_i.$$

Next, an attention operator is used to aggregate the motif encoding using $\boldsymbol{h}_{G_i}$ as query and $\boldsymbol{h}_{m_i}$s as keys and values. The resulting encoding, denoted as $\boldsymbol{h}'_{G_i}$, is the aggregated molecular encoding by

combining motifs in $M_i$.

$$e_{G_i m_i} = g(\boldsymbol{W}\boldsymbol{h}_{G_i}, \boldsymbol{W}\boldsymbol{h}_{m_i}), \alpha_{G_i m_i} = \frac{\exp(e_{G_i m_i})}{\sum_{m_i \in M_i} \exp(e_{G_i m_i})}, \boldsymbol{h}'_{G_i} = \sum_{m_i \in M_i} \alpha_{G_i m_i} \cdot \boldsymbol{h}_{m_i},$$

where $g(\cdot)$ is an attention operator, and $\boldsymbol{W}$ is weight matrix. The target classifier $f(\cdot)$ then uses aggregated encoding to generate predictions. The training process is designed to minimize the cross-entropy loss between the predicted probabilities of the aggregated molecular encoding and those of the original molecular encoding:

$$\mathcal{L} = \sum_i \mathcal{L}_{CE}(f(\boldsymbol{h}_{G_i}), f(\boldsymbol{h}'_{G_i})). \tag{1}$$

The attention operator is trained to identify important motifs to represent the molecule. This leads to the molecule-motif relation score $\boldsymbol{\alpha}_{G_i m_i}$ and the probabilities associated with label predictions $\hat{\boldsymbol{y}} = f(\boldsymbol{h}'_{G_i})$.

**Stage 2: class-motif relation score calculation.**
We repeat the process of Stage 1 to compute the molecule-motif scores for each molecular graph. After that, we create a molecule-motif score matrix $\boldsymbol{S}^{mm} \in \mathbb{R}^{V \times |\mathcal{G}|}$ and a molecule-class probability matrix $\boldsymbol{P} \in \mathbb{R}^{|\mathcal{G}| \times C}$. Here, $S^{mm}_{pk}$ represents the molecule-motif relation score between molecule $k$ and motif $p$. If the motif $i$ does not appear in the molecule $j$, we set $S^{mm}_{ij} = 0$. Element $P_{kr}$ denotes the probability that molecule $k$ is associated with label $r$.

Subsequently, we perform a matrix multiplication between $\boldsymbol{S}^{mm}$ and $\boldsymbol{P}$ to get class-motif relation score matrix $\boldsymbol{S}^{cm}$:

$$\boldsymbol{S}^{cm} = \boldsymbol{S}^{mm}\boldsymbol{P}, S^{cm}_{pr} = \sum_{k=1}^{|\mathcal{G}|} S^{mm}_{pk} P_{kr}$$

which represents the relationship between motif $p$ and class $r$. Then, we normalize $\boldsymbol{S}^{cm}$ by dividing each column, corresponding to a motif, by the number of molecules it appears.

**Stage 3: class-wise motif filtering.**
We filter motifs for each class using class-wise motif scores from the previous stage. The motif set for the class $C_r$ is

$$M_{C_r} = \{m_i | \boldsymbol{S}^{cm}_{ir} > \theta, 1 \le i \le V\} \tag{2}$$

where $\theta$ is a hyper-parameter to control the size of the important motif vocabulary. An illustration of our class-wise motif identification is provided in Figure 2.

## 3.3 Class-wise Motif-based Graph Generation

Section 3.2 identifies motifs of high relevance for each class. Building on this, our methodology employs a motif-based generative model to construct model-level explanations tailored to each class. The motif-based molecular generation method can be divided into VAE-based (Jin et al., 2018; 2020) and RL-based methods (Yang et al., 2021). In our approach, we employ a VAE-based method as a generator. However, our approach can easily adapt to RL-based methods. The graph generation process is illustrated in Figure 3.

### 3.3.1 Tree Decomposition

Given a target class $r$, a set of important motifs $M_{C_r}$ associated with the class $r$, and the set of molecules $\mathcal{G}_r$ whose predicted class by trained GNN is $r$. For each molecular graph $G_i \in \mathcal{G}_r$, we first identify the motifs in $M_{C_r}$ from $G_i$: $M_i^r = M_i \cap M_{C_r}$. Subsequently, we pinpoint every edge in $G_i$ that does not belong to any motif in $M_i^r$. We treat each motif in $M_i^r$ and each non-motif edge as a cluster node. Then, we construct a cluster graph by connecting clusters with intersections. Finally, a spanning tree $\mathcal{T}$ is selected from this cluster graph, defined as the motif junction tree for $G_i$.

Figure 3: Class-wise motif-based graph generation. Starting with a molecular graph, we first construct a junction tree. Next, a tree encoder is applied to obtain a tree encoding, which is then decoded by a tree decoder to reconstruct the junction tree. Finally, a graph decoder uses the predicted junction tree to reproduce the molecular graph.

### 3.3.2 GRAPH ENCODER

We encode the latent representation of $G = (\boldsymbol{A}, \boldsymbol{X})$ by the target GNN feature extractor $\phi(\cdot)$: $\boldsymbol{h}_G = \phi(\boldsymbol{A}, \boldsymbol{X})$. The mean $\boldsymbol{\mu}_G$ and log variance $\log \boldsymbol{\sigma}_G$ of the variational posterior approximation are computed from $\boldsymbol{h}_G$ with two separate affine layers. $\boldsymbol{z}_G$ is sampled from a Gaussian distribution $\mathcal{N}(\boldsymbol{\mu}_G, \boldsymbol{\sigma}_G)$. In the variational posterior approximation, we use two different affine layers to compute the mean, represented as $\boldsymbol{\mu}_G$, and the log variance, denoted as $\log \boldsymbol{\sigma}_G$, from $\boldsymbol{h}_G$. Then, we sample $\boldsymbol{z}_G$ from a Gaussian distribution $\mathcal{N}(\boldsymbol{\mu}_G, \boldsymbol{\sigma}_G)$.

### 3.3.3 TREE ENCODER

We employ a graph neural network to encode the junction tree $\mathcal{T} = (\boldsymbol{A}_\mathcal{T}, \boldsymbol{X}_\mathcal{T})$, where every node $i$'s feature $x_i$ is initially encoded by feeding its corresponding motif graph to $\phi(\cdot)$. Formally, the node embedding and tree embedding are updated as follows:

$$\boldsymbol{H}_\mathcal{T} = \mathrm{GNN}(\boldsymbol{A}_\mathcal{T}, \boldsymbol{X}_\mathcal{T}), \boldsymbol{h}_\mathcal{T} = \mathrm{AGG}(\boldsymbol{H}_\mathcal{T}).$$

where $\mathrm{GNN}(\cdot)$ is a graph neural network, and $\mathrm{AGG}(\cdot)$ is an aggregation function. $\boldsymbol{z}_\mathcal{T}$ is sampled similarly as in the graph encoder, where $\boldsymbol{z}_\mathcal{T}$ is a tree encoding.

### 3.3.4 TREE DECODER

A tree decoder is used to decode a junction tree $\mathcal{T}$ from a tree encoding $\boldsymbol{z}_\mathcal{T}$. The decoder builds the tree in a top-down approach, where nodes are created sequentially, one after the other. The node under consideration first obtains its embedding by inputting the existing tree into the tree encoder, and the resulting output is then used as the node embedding. The node embedding is utilized for two predictions: determining whether the node has a child and identifying the child's label if a child exists. Formally, for node $i$

$$\boldsymbol{H}' = \mathrm{GNN}(\boldsymbol{A}', \boldsymbol{X}'), \boldsymbol{p}_i = \mathrm{PRED}(\mathrm{COMB}(\boldsymbol{z}_\mathcal{T}, \boldsymbol{H}_i')), \boldsymbol{q}_i = \mathrm{PRED}^l(\mathrm{COMB}^l(\boldsymbol{z}_\mathcal{T}, \boldsymbol{H}_i'))$$

where $\boldsymbol{A}'$ and $\boldsymbol{X}'$ are the existing tree's adjacent matrix and node features, respectively. $\mathrm{COMB}(\cdot)$ and $\mathrm{COMB}^l(\cdot)$ are combination functions, and $\mathrm{PRED}(\cdot)$ and $\mathrm{PRED}^l(\cdot)$ are predictors. When the current node does not have a new child, the decoder will go to the next node in the existing tree.

### 3.3.5 GRAPH DECODER

The final step is reproducing the molecular graph from the predicted junction tree $\mathcal{T}$. Let $\mathcal{G}(\mathcal{T})$ be a set of graphs whose junction tree is $\mathcal{T}$. The decoder aims to find $\hat{G}$ as

$$\hat{G} = \arg \max_{G' \in \mathcal{G}(\mathcal{T})} f^a(G').$$

In our approach, we aim to ensure that the reproduced molecule maintains the characteristics of the target GNN. So we design the score function as $f^a(G_i) = f(\phi(G_i))[r]$, where $r$ is the target label.

### 3.3.6 LOSS FUNCTION

The loss function for graph generation comprises two components: the reconstruction loss and the property loss. The reconstruction loss ensures the generated graph matches the distribution of the original dataset. The property loss ensures the behavior of the generated graph aligns with the target GNN.

The reconstruction loss can be formally defined as below:

$$\mathcal{L}_\mathcal{R} = \sum \mathcal{L}_{child} + \sum \mathcal{L}_{label} \tag{3}$$

, where $\mathcal{L}_{child}$ corresponds to predicting whether a node has a child, while $\mathcal{L}_{label}$ represents the loss associated with predicting the label of the newly added child. We perform teacher forcing for the reconstruction loss to make the information get the correct history at each step.

The property loss is like below:

$$\mathcal{L}_\mathcal{P} = \sum \mathrm{MSE}(\boldsymbol{h}_G, \boldsymbol{h}_\mathcal{T}) + \sum \mathcal{L}_{CE}(f(\boldsymbol{h}_\mathcal{T}), r) \tag{4}$$

Table 1: Validity results on six real-world datasets. OOM refers to out of memory.

| Datasets | Models | Label 0 | Label 1 | Average |
|---|---|---|---|---|
| Mutagenicity | XGNN | 0.34 | 0.25 | 0.295 |
| | GNNInterpreter | 0.11 | 0.21 | 0.16 |
| | **Ours** | 1.00 | 1.00 | 1.00 |
| PTC_MR | XGNN | 0.70 | 0.21 | 0.45 |
| | GNNInterpreter | 0.10 | 0.05 | 0.07 |
| | **Ours** | 1.00 | 1.00 | 1.00 |
| PTC_MM | XGNN | 0.49 | 0.56 | 0.51 |
| | GNNInterpreter | 0.15 | 0.20 | 0.17 |
| | **Ours** | 1.00 | 1.00 | 1.00 |
| PTC_FM | XGNN | 0.48 | 0.37 | 0.42 |
| | GNNInterpreter | 0.13 | 0.08 | 0.10 |
| | **Ours** | 1.00 | 1.00 | 1.00 |
| AIDS | XGNN | 0.92 | 0.71 | 0.81 |
| | GNNInterpreter | 0.09 | 0.08 | 0.08 |
| | **Ours** | 1.00 | 1.00 | 1.00 |
| NCI-H23 | XGNN | OOM | OOM | OOM |
| | GNNInterpreter | 0.59 | 0.62 | 0.60 |
| | **Ours** | 1.00 | 1.00 | 1.00 |

, where $h_{\mathcal{T}}$ is the embedding of the generated tree, $\mathcal{T}$ represents the predicted motif-node tree, $h_G$ is the corresponding graph embedding, $G$ refers to the generated molecular graph, $f(\cdot)$ is the target classifier, and $r$ is the target label. The final loss function is

$$\mathcal{L} = \mathcal{L}_{\mathcal{R}} + \mathcal{L}_{\mathcal{P}} \tag{5}$$

### 3.3.7 EXPLANATION GENERATION AT SAMPLING TIME

During the sampling process, the method begins by randomly sampling a tree encoding $z_{\mathcal{T}}$, This encoding is then passed through a tree decoder to reconstruct a junction tree $\mathcal{T}$. Subsequently, a graph decoder generates the explanation based on the predicted junction tree $\mathcal{T}$. The whole process is shown in Figure 3.3.7.

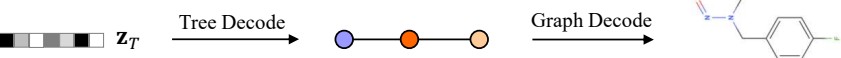

Figure 4: An illustration of the explanation sampling process.

## 4 EXPERIMENT

We conduct qualitative and quantitative experiments on six real-world datasets to evaluate the effectiveness of our proposed methods.

### 4.1 DATASET AND EXPERIMENTAL SETUP

We evaluate the proposed methods using molecule classification tasks on six real-world datasets: Mutagenicity, PTC_MR, PTC_MM, PTC_FM, AIDS, and NCI-H23. The details of six datasets and experimental settings are represented in Appendix B and C.

**Baselines.** We compare our MAGE model with two state-of-the-art baselines: XGNN and GNNInterpreter. Noted that all methods are tested using official implementation and compared in a fair setting. We extract motifs by identifying "bridge bonds". We build a variant that uses the same motif extraction and graph generation processes but without our attention-based motif identification step. This variant is to demonstrate the effectiveness of our class-wise motif identification process. The details of this variant will be discussed in section 4.2.

**Evaluation metrics.** Our quantitative evaluation uses two distinct metrics to assess performance: Validity and Average Probability.

Table 2: Quantitative results for different explanation methods. We highlight the average class probability (higher is better) for six datasets. The best performances on each dataset are shown in **bold**. OOM refers to out of memory.

| Datasets | Models | Label 0 | Label 1 | Average |
|---|---|---|---|---|
| Mutagenicity | XGNN | $0.8992 \pm 0.0835$ | $0.9831 \pm 0.0514$ | 0.9411 |
| | GNNInterpreter | $0.8542 \pm 0.3198$ | $0.9938 \pm 0.0191$ | 0.9240 |
| | Variant | $0.9897 \pm 0.0754$ | $0.9888 \pm 0.0302$ | 0.9892 |
| | **Ours** | $\mathbf{0.9977 \pm 0.0032}$ | $\mathbf{0.9941 \pm 0.0240}$ | **0.9959** |
| PTC_MR | XGNN | $0.9906 \pm 0.0718$ | $0.9698 \pm 0.0417$ | 0.9802 |
| | GNNInterpreter | $0.9067 \pm 0.1728$ | $0.9697 \pm 0.0211$ | 0.9382 |
| | Variant | $0.9902 \pm 0.0480$ | $0.9641 \pm 0.1163$ | 0.9771 |
| | **Ours** | $\mathbf{0.9961 \pm 0.0375}$ | $\mathbf{0.9918 \pm 0.0621}$ | **0.9939** |
| PTC_MM | XGNN | $0.9899 \pm 0.0994$ | $0.9266 \pm 0.1059$ | 0.9582 |
| | GNNInterpreter | $0.9601 \pm 0.0638$ | $0.9541 \pm 0.0207$ | 0.9571 |
| | Variant | $0.9784 \pm 0.0475$ | $0.9693 \pm 0.0279$ | 0.9738 |
| | **Ours** | $\mathbf{0.9914 \pm 0.0342}$ | $\mathbf{0.9833 \pm 0.0019}$ | **0.9873** |
| PTC_FM | XGNN | $0.9967 \pm 0.0309$ | $0.9380 \pm 0.0991$ | 0.9673 |
| | GNNInterpreter | $0.9945 \pm 0.0147$ | $0.9460 \pm 0.0295$ | 0.9702 |
| | Variant | $0.9882 \pm 0.0024$ | $0.9790 \pm 0.0024$ | 0.9836 |
| | **Ours** | $\mathbf{0.9979 \pm 0.0024}$ | $\mathbf{0.9890 \pm 0.0024}$ | **0.9934** |
| AIDS | XGNN | $0.9259 \pm 0.1861$ | $\mathbf{0.9977 \pm 0.0225}$ | 0.9618 |
| | GNNInterpreter | $0.4600 \pm 0.4983$ | $0.9973 \pm 0.0116$ | 0.7286 |
| | Variant | $0.9802 \pm 0.1112$ | $0.9939 \pm 0.0564$ | 0.9870 |
| | **Ours** | $\mathbf{0.9883 \pm 0.0663}$ | $0.9903 \pm 0.0539$ | **0.9893** |
| NCI-H23 | XGNN | OOM | OOM | OOM |
| | GNNInterpreter | $0.9883 \pm 0.0711$ | $\mathbf{0.9997 \pm 0.0015}$ | **0.9940** |
| | Variant | $0.9874 \pm 0.0685$ | $0.9863 \pm 0.01069$ | 0.9868 |
| | **Ours** | $\mathbf{0.9936 \pm 0.0329}$ | $0.9934 \pm 0.0422$ | 0.9935 |

*Validity.* This metric is defined as the proportion of chemically valid molecules out of the total number of generated molecules (Bilodeau et al., 2022).

$$\text{Validity} = \frac{\# \text{ valid molecules}}{\# \text{ generated molecules}}$$

This metric serves as a critical indicator of the practical applicability of our method, ensuring that the generated molecules not only align with the intended class but also adhere to fundamental chemical validity criteria.

*Average Probability.* Following the GNNInterpreter (Wang & Shen, 2022), this metric calculates the average class probability and the standard deviation of these probabilities across 1000 explanation graphs for each class within all six datasets.

$$\text{Average Probability} = \frac{\sum \text{class probability}}{\# \text{ generated molecules}}.$$

This approach provides a comprehensive measure of the consistency and reliability of the explanation graphs in representing different classes.

Together, these two metrics offer a robust framework for evaluating the effectiveness of our approach from both the accuracy and applicability perspectives. Considering XGNN's limitation in generating explanation graphs with edge features, we assign the default bond type from the RDKit library as the default edge feature for explanations produced by XGNN. We also report the training time and sampling time of different models in appendix E.

## 4.2 QUANTITATIVE RESULTS

Table 1 details the Validity results for various models across six datasets. A notable observation from these results is that our method consistently produces valid molecular explanation graphs for both classes in all cases. This reliability stems from our method's use of valid substructures as foundational elements, effectively overcoming the limitation seen in atom-by-atom generation methods, which often compel the model to create chemically invalid intermediate structures.

Table 3: The qualitative results for Mutagenicity dataset. The first class row of the table displays the explanation graphs corresponding to the Nonmutagen label, while the second class row presents the explanation graphs for the Mutagen label. The final column in the table provides examples of actual graphs. The different colors in the nodes represent different values in the node feature

| Class | Generated Model-Level Explanation Graphs for Mutagenicity Dataset | | | |
|---|---|---|---|---|
| | XGNN | GNNInterpreter | Ours | Example |
| Nonmutagen |  |  |  |  |
| |  |  |  |  |
| |  |  |  |  |
| |  |  |  |  |
| Mutagen |  |  |  |  |
| |  |  |  |  |

Additionally, the data reveals that XGNN exhibits better validity than GNNInterpreter. The reason for this difference lies in the design of the algorithms: XGNN incorporates a manual constraint on the maximum atom degree, which aids in maintaining chemical validity. In contrast, GNNInterpreter is to align the embedding of its generated explanations with the average embedding of the dataset, which does not inherently ensure chemical validity and leads to more invalid generated molecules.

These findings from the validity experiment highlight a principle: maximizing the inductive biases inherent in the molecular graph structure is important. It enables models to achieve higher levels of validity in the outputs, demonstrating that careful consideration of the structural characteristics of molecules can significantly enhance the performance and reliability of these models.

Table 2 shows the average probability of four models on six real-world datasets. Our method outperforms the baseline methods in ten out of twelve experiments, clearly highlighting its effectiveness. Additionally, our method consistently delivers strong results across both classes in each dataset. This contrasts XGNN and GNNInterpreter, which tend to excel in one class but fall short in the other. We attribute this success of our method to its unique strategy of generating explanations in a motif-by-motif way, effectively reducing the influence of extraneous or noise atoms in the data.

Furthermore, we conducted a comparative analysis between our method and a variant model. This variant model operates by first identifying all potential motifs within the dataset. Once these motifs are extracted, it assesses them by putting each motif as an input to the target model. The evaluation of each motif is based on the output probability associated with a particular class, which is assigned as the motif's score. Following this scoring process, the method ranks all motifs in order of their scores. Only motifs whose scores exceed 0.9 are chosen to refine the selection.

The outcomes of our study indicate that our method outperforms the variant model across all classes. This significant improvement underscores the efficiency of our attention-based motif learning module. Unlike the variant model, which solely focuses on the individual motifs without considering their functional roles within the molecules, our method learns and understands the interplay between motifs and molecules. This enables our approach to effectively identify and select motifs relevant to the specific molecular context while simultaneously filtering out motifs that do not contribute meaningfully, often called noise motifs.

## 4.3    QUALITATIVE RESULTS

Table 3 presents the qualitative assessment of three different methods applied to the two classes of the Mutagenicity dataset. The table also includes actual graph examples from the dataset for

reference. Our analysis reveals that our method can consistently produce human-understandable and valid explanation graphs, which XGNN and GNNInterpreter seem to lack. More qualitative results can be found in Appendix I.

We observed a trend in GNNInterpreter to produce explanations that are disconnected. This is likely due to its approach of aligning explanation embeddings with the dataset's average embedding, resulting in a focus on node features rather than structural details. XGNN, on the other hand, tends to create explanations with invalid ring structures. Although XGNN imposes constraints on the degree of each atom, its atom-by-atom generation approach makes it challenging to form valid ring structures. Our method can generate explanations with meaningful substructures like rings.

This inability of XGNN and GNNInterpreter to effectively identify complex molecular structures may contribute to their lower validity in explanation generation. Our method, in contrast, successfully navigates these complexities, offering more meaningful, accurate and structurally coherent explanations. This difference highlights the importance of considering both node features and molecular structures in the generation of explanation graphs, a balance that our approach seems to strike effectively.

### 4.4 ABLATION STUDY: LOSS FUCTION

Table 4: Comparison of different components of the loss function on Mutagenicity dataset.

| Loss function | Label 0 | Label 1 |
|:---:|:---:|:---:|
| $\mathcal{L}_R$ | $0.9827 \pm 0.0580$ | $0.9892 \pm 0.0482$ |
| $\mathcal{L}_P$ | $0.9881 \pm 0.0518$ | $0.9805 \pm 0.0589$ |
| $\mathcal{L}_R + \mathcal{L}_P$ | $0.9977 \pm 0.0032$ | $0.9941 \pm 0.0240$ |

In this section, we evaluate performance on Mutagenicity dataset when using each loss component independently. Table 4 shows the results. From the table, we observe that using only the reconstruction loss results in high accuracy, though it is slightly lower than configurations that include property alignment. This indicates that, while the reconstruction loss is proficient at preserving input structure, it may lack alignment with target-specific characteristics. The table also shows that using only the property loss results in slightly lower performance than the combined loss. This suggests that, without the reconstruction loss, the model may lose some structural accuracy, leading to a minor decrease in overall performance.

## 5 RELATED WORK

Model-level explanation methods in the field are currently under-researched, with only a few studies addressing this issue. These methods can be divided into two categories. The first category is concept-based methods. GCExplainer (Magister et al., 2021) introduced the incorporation of concepts into GNN explanations, using the k-Means clustering algorithm on GNN embeddings to identify clusters, each representing a concept. PAGE (Shin et al., 2022) provides clustering on the embedding space to discover propotypes as explanations. GLGExplainer (Azzolin et al., 2022) adopts prototype learning to identify data prototypes and then uses these in an E-LEN model to create a Boolean formula that replicates the GNN's behavior. GCNeuron (Xuanyuan et al., 2023), on the other hand, employs human-defined rule in natural language and considers graphs with certain masked nodes as concepts, using compositional concepts with the highest scores for global explanations. The second category comprises generation-based methods. XGNN (Yuan et al., 2020) uses deep reinforcement learning to generate explanation graphs node by node. GNNInterpreter (Wang & Shen, 2022), alternatively, learns a probabilistic model that identifies the most discriminative graph patterns for explanations.

## 6 CONCLUSION

This work proposes a novel motif-based model-level explanation method for Graph Neural Networks, specifically applied to molecular datasets. Initially, the method employs a motif extraction technique to identify all potential motifs. Following this, a one-layer attention mechanism is used to evaluate the relevance of each motif, filtering out those that are not pertinent. Finally, a graph generator is trained to produce explanations based on the chosen motifs. Compared to two baseline methods, XGNN and GNNInterpreter, our proposed method offers explanations that are more human-understandable, accurate, and structurally consistent.

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

## A    DETAILS OF DECODE A VALID MOLECULE EXPLANATION

To decode a valid molecule explanation from a tree structure, we have the following steps:

1. Start with a decoded Tree: The decoding process begins with a decoded tree $\mathcal{T}$, which represents the hierarchical structure of motifs.

2. Enumerate motif Combinations: For each motif node in the tree and its neighboring motifs, we enumerate different combinations to find feasible arrangements. Certain combinations lead to chemically infeasible molecules, and are discarded from further consideration.

3. Rank and Combine combined subgraphs: At each node, we rank the combined subgraphs based on our target GNN, ensuring that the final graph is following the same distribution of our target class.

4. The final graph is decoded by putting together all the predicted subgraphs.

---

**Algorithm 1** Decoding a Molecule from a Tree Structure

---

**Require:** Decoded tree $\mathcal{T}$ representing the hierarchical structure of motifs.
**Ensure:** Final molecular graph $G$.
 1: **Initialization**: Set the final graph $G$ to an empty graph.
 2: **for** each motif node $m$ in the tree $\mathcal{T}$ **do**
 3:      Identify neighboring motifs of $m$ in $\mathcal{T}$.
 4:      Enumerate all feasible combinations of $m$ with its neighboring motifs.
 5:      Discard combinations that lead to chemically infeasible molecules.
 6:      Rank the remaining combined subgraphs using the target GNN model to assess alignment with the target class distribution.
 7:      Select the top-ranked combined subgraph.
 8:      Integrate the selected subgraph into the final graph $G$.
 9: **end for**
10: **Output**: The final molecular graph $G$ obtained by assembling all predicted subgraphs.

---

Algorithm shows the details of our graph decoder.

## B    DETAILS OF DATASETS

Table 5: Statistics and properties of four real-world molecule datasets.

|  | **Mutag** | **PTC_MR** | **PTC_MM** | **PTC_FM** | **AIDS** | **NCI-H23** |
|---|---|---|---|---|---|---|
| # Nodes (avg) | 30.32 | 14.29 | 14.32 | 14.48 | 15.69 | 26.07 |
| # Edges (avg) | 30.77 | 14.69 | 13.97 | 14.11 | 16.20 | 28.10 |
| # Graphs | 4337 | 344 | 336 | 349 | 2000 | 40353 |
| # Classes | 2 | 2 | 2 | 2 | 2 | 2 |

The statistics and properties of four datasets are summarized in Table 5.

Mutagenicity (Kazius et al., 2005; Riesen & Bunke, 2008) is a chemical compound dataset containing 4,337 molecule graphs. Each of these graphs is classified into either mutagen or non-mutagen categories, signifying their mutagenic impact on the Gram-negative bacterium Salmonella typhimurium.

PTC_MR, PTC_MM, and PTC_FM (Morris et al., 2020) are three collections of chemical compounds, each respectively documenting carcinogenic effects on male rats (MR), male mice (MM), and female mice (FM).

AIDS (Morris et al., 2020) comprises 2,000 graphs that represent molecular compounds, all of which are derived from the AIDS Antiviral Screen Database of Active Compounds.

NCI-H23 (Morris et al., 2020) is a collection of 40353 molecular compounds to evaluate the efficacy against lung cancer cells.

## C  EXPERIMENTAL SETTINGS

By following baseline approaches, our experiments adopt a simple GCN model and focus on explanation results. For the target GNN, we use a 3-layer GCN as a feature extractor and a 2-layer MLP as a classifier on all datasets. The model is pre-trained and achieves reasonable performances on all datasets. Following the (Xu et al., 2018), the hidden dimension is selected from $\{16, 64\}$. We employ mean-pooling as the readout function and ReLU as the activation function. We use Adam optimizer for training. The target model is trained for 100 epochs, and the learning rate is set to 0.01. The CPU in our setup is an AMD Ryzen Threadripper 2990WX, accompanied by 256 GB of memory, and the GPU is an RTX 4090. For training an explainer, the number of epochs is selected from $\{50, 100\}$, the learning rate is selected from $\{0.01, 0.0001, 0.0005\}$, the batch size is selected from $\{32, 320\}$.

## D  STUDY OF MOTIF EXTRACTION TIME

In this section, we study the motif extraction time for all six datasets.

|  | Average Number of Nodes | Motif Extraction Time (ms/graph) |
|---|---|---|
| Mutagenicity | 30.23 | 0.799 |
| PTC_MR | 14.29 | 0.428 |
| PTC_MM | 13.97 | 0.415 |
| PTC_FM | 14.11 | 0.444 |
| AIDS | 15.69 | 0.457 |
| NCI-H23 | 26.07 | 0.877 |

The table demonstrates that the bridge bond cut-off method achieves decomposition times in the millisecond range. This approach allows us to process large datasets with minimal computational cost.

## E  STUDY OF TRAINING AND INFERENCE TIME

We also present a detailed comparison of the training and sampling times between our method and other baseline approaches in table 6. Here, sampling time is the average time to generate a single explanation. Compared to the two baselines, a key distinction of our method lies in its requirement for an initial training phase for the explanation graph generator. This phase confers a significant benefit: the ability to efficiently sample explanations. Our experimental data underscores this advantage. Following the completion of the training phase for the explanation generator, our method demonstrated the capacity to generate a large number of explanations rapidly. Specifically, it could sample 1000 explanations in a matter of seconds. This starkly contrasts other baseline methods, which required at least 3.3 hours to produce the same number of examples. This dramatic reduction in sampling time not only highlights the efficiency of our method but also suggests its practical applicability in scenarios where quick generation of explanations is crucial. Due to the space limit, we put the study of hyper-parameter $\theta$ in Appendix F.

## F  STUDY OF HYPERPARAMETER

Table 7 shows the study of hyperparameter $\theta$, we found that as $\theta$ increases, the number of motifs selected for the Mutagen class decreases. The average probability rises until $\theta$ reaches 0.10, where peak performance is observed, after which it starts to decline. This trend occurs because when $\theta$ is low, some irrelevant motifs are included in the explanations. As $\theta$ increases, these 'noise' motifs

Table 6: Training and inference time for different explanation methods. XGNN and GNNInterpreter do not have training time because they only have sampling process.

| Datasets | Models | Training Time | Sampling Time |
|---|---|---|---|
| Mutagenicity | XGNN | — | 54s |
| | GNNInterpreter | — | 29s |
| | Variant | 528s | 0.0015s |
| | **Ours** | 539s | 0.0029s |
| PTC_MR | XGNN | — | 62s |
| | GNNInterpreter | — | 27s |
| | Variant | 338s | 0.0005s |
| | **Ours** | 368s | 0.0006s |
| PTC_MM | XGNN | — | 46s |
| | GNNInterpreter | — | 17s |
| | Variant | 303s | 0.0023s |
| | **Ours** | 367s | 0.0002s |
| PTC_FM | XGNN | — | 47s |
| | GNNInterpreter | — | 12s |
| | Variant | 307s | 0.0027s |
| | **Ours** | 386s | 0.0018s |
| AIDS | XGNN | — | 19s |
| | GNNInterpreter | — | 32s |
| | Variant | 468s | 0.0065s |
| | **Ours** | 513s | 0.0076s |
| NCI-H23 | XGNN | — | OOM |
| | GNNInterpreter | — | 21s |
| | Variant | 10189s | 0.0115s |
| | **Ours** | 13657s | 0.0126s |

Table 7: Study of Hyperparameter $\theta$ on Mutagen class of Mutagenicity dataset

| | $\theta = 0.01$ | $\theta = 0.05$ | $\theta = 0.10$ | $\theta = 0.20$ |
|---|---|---|---|---|
| # Selected Motifs | 1591 | 1328 | 1206 | 1073 |
| Average Probability | $0.9621 \pm 0.1151$ | $0.9850 \pm 0.0784$ | $0.9977 \pm 0.0032$ | $0.9753 \pm 0.0981$ |

are filtered out, enhancing performance. However, if $\theta$ is set too high, the resulting small number of selected motifs is insufficient for forming a comprehensive explanation.

## G    STUDY OF COMPARISON BETWEEN INSTANCE-LEVEL EXPLAINERS AND MODEL-LEVEL EXPLAINER

In this section, we compare instance-level explanation methods with our proposed method on Mutagenicity dataset. To address this, we performed an experiment where we computed the Average Probability of two local explainers, GNNExplainer (Ying et al., 2019) and GEM (Lin et al., 2021), by averaging their probability outputs across various instances, as per your suggestion. The results for GNNExplainer were obtained using the PyTorch Geometric library, while GEM results were produced with its official implementation. The table below presents the findings.

From the results, it is evident that our MAGE approach outperforms the instance-level explainers significantly. This demonstrates the advantage of generating explanations from a global (model-level) perspective as opposed to relying on a local (instance-level) perspective. Model-level explanations generate insights leveraging global patterns and the overall model behavior, whereas instance-level explanations focus on generating explanations for individual data points or molecules. This distinction

Table 8: Comparison of instance-level explainer and MAGE

| Methods | Label 0 | Label 1 |
|---|---|---|
| GNNExplainer | $0.8339 \pm 0.1404$ | $0.8250 \pm 0.1402$ |
| GEM | $0.8739 \pm 0.1428$ | $0.8887 \pm 0.1306$ |
| MAGE(Ours) | $0.9977 \pm 0.0032$ | $0.9941 \pm 0.0240$ |

allows model-level explanations like MAGE to capture broader, more generalizable patterns within the dataset, which can provide a more holistic understanding of the model's decision-making process. Conversely, instance-level methods such as GNNExplainer and GEM may provide more localized and specific insights, but often at the expense of lacking global context.

## H  STUDY OF NOVELTY OF OUR GENERATED EXPLANATION

To further validate that our explainer does not simply memorize the training set, we tested the outputs of our explainer using the novelty metric, which is widely used in the graph generation field. The novelty metric measures the percentage of generated graphs that do not overlap with the training set, providing an effective way to assess whether the generator is simply reproducing seen examples.

Table 9: Study of novelty of generated explanations on all six datasets

| Datasets | Novelty of Label 0 | Novelty of Label 1 |
|---|---|---|
| Mutagenicity | 1.0 | 1.0 |
| PTC_MR | 1.0 | 1.0 |
| PTC_MM | 1.0 | 1.0 |
| PTC_FM | 1.0 | 1.0 |
| AIDS | 1.0 | 1.0 |
| NCI-H23 | 1.0 | 1.0 |

Our results in the table 9 demonstrate that our method achieves a 100% novelty score across all six datasets, indicating that the generated explanations are not merely memorized training examples but reflect the learned properties of the target GNN.

These findings further support that our framework generates meaningful and novel explanations aligned with the target GNN, rather than relying on trivial reproduction of training data. We have added this part in Appendix in our revised paper.

# I   MORE QUALITATIVE RESULTS

Table 10 and table 11 are additional qualitative results of explanations on Mutagenicity dataset. Each samples are randomly selected from the generated molecular graphs.

Table 10: More qualitative results for Nonmutagen class on Mutagenicity dataset.

Table 11: More qualitative results for Mutagen class on Mutagenicity dataset.

| Class | XGNN | GNNInterpreter | Ours | Example |
|-------|------|----------------|------|---------|
| Mutagen | | | | |

