# OpenReview forum: "MAGE: Model-Level Graph Neural Networks Explanations via Motif-based Graph Generation"
_ICLR.cc/2025/Conference — ICLR 2025 Poster_

### Official Review · Reviewer_9aZJ · 2024-11-01

**Soundness:** 2
**Presentation:** 1
**Contribution:** 2
**Rating:** 6
**Confidence:** 4

**Summary:**

This paper introduces MAGE (Motif-bAsed GNN Explainer), which improves interpretability by using motifs as core units in explanations. MAGE identifies class-specific motifs through decomposition and attention-based learning, creating clearer molecular graph explanations. This approach, validated on six datasets, provides more human-understandable results. However, there are some issues in this article that need clarification.

**Strengths:**

1. This paper introduces a new method, MAGE, to generate motif-based graph explanations.
2. This paper is well-organized, it is easy to follow the main idea.
3. This paper conducts lots of experiments to verify the effectiveness of this method.

**Weaknesses:**

1. Some symbols are confusing, such as $\mathcal{L}_T$,$\mathcal{L}_L$. It seems a trivial solution exists that the $G$ is the same as the $\mathcal{T}$ from the loss function.
2. The figures are not well expressed. For example, in Figure 3, there are two graph decoders. However, from the paper, I can only find one graph decoder.
3. It is confusing how to construct a new graph from different subgraphs. It's better to give the algorithm instructions on how to construct a new graph from subgraphs and how they share nodes.

Some minor suggestions.
1. Notations are inconsistent.  $\mathbf{A}$ and $A$ are used interchangablelly.
2.  It is confusing to use mathbf{} to denote matrix (A) and set (V)
3.  $N$ is used with different meanings. "N represents the total number of atoms in the graph" & "Given a dataset, denoted as G with N molecules and C classes"
4.  Line 271, z_T is not defined.
5. Format issue, cite should be \citep{}

**Questions:**

1. How to a construct graph from subgraphs?  Do the subgraphs share some nodes?

---

> ### Author Response · Authors · 2024-11-15
> **Response to Reviewer 9aZJ**
>
> We sincerely thank the reviewer for the thorough and insightful review.
>
> Our responses to all questions are below:
>
> Q1. **Some symbols are confusing, such as $\mathcal{L}_T$, $\mathcal{L}_L$. It seems a trivial solution exists that the $G$ is the same as the $T$ from the loss function.**
>
> Answer: Thank you for your question and for pointing out the potential confusion with the symbols. To enhance clarity, we have updated our notation:
>
> $\mathcal{L}_{child}$ now represents the loss term associated with predicting whether a node has a child.
>
> $\mathcal{L}_{label}$ represents the loss associated with predicting the label of the newly added child.
>
> Clarification of Symbols:
>
> $G$: Refers to the generated molecular graph.
>
> $\mathcal{T}$: Represents the predicted motif-node tree, which captures the structural relationships among motifs.
>
> We hope this clarification of symbols helps, and we appreciate your feedback on enhancing clarity around these terms. We have revised the notation and adding explanatory text to make these distinctions clearer in the paper. (line 316-320, 345-347)
>
> Q2. **The figures are not well expressed. For example, in Figure 3, there are two graph decoders. However, from the paper, I can only find one graph decoder.**
>
> Answer: Thank you for pointing this out. We apologize for any confusion caused by Figure 3. To clarify, there is indeed only one graph decoder in our model, as stated in the paper. We have revised the description of Figure 3 to more accurately depict the role of the single graph decoder and avoid any misinterpretation.
>
> Q3. **How to a construct graph from subgraphs? Do the subgraphs share some nodes?**
>
> Answer: Thank you for the question. We use a similar method to the [1], adapting it to decode a valid molecule explanation from a tree structure. This approach ensures chemical feasibility and structural integrity in the final molecular graph. Here’s an overview of the steps:
>
> 1. Start with a decoded Tree: The decoding process begins with a decoded tree $\mathcal{T}$, which represents the hierarchical structure of motifs.
>
> 2. Enumerate motif Combinations: For each motif node in the tree and its neighboring motifs, we enumerate different combinations to find feasible arrangements. Certain combinations lead to chemically infeasible molecules, and are discarded from further consideration.
>
> 3. Rank and Combine combined subgraphs: At each node, we rank the combined subgraphs based on our target GNN, ensuring that the final graph is following the same distribution of our target class.
>
> 4. The final graph is decoded by putting together all the predicted subgraphs.
>
> By following this structured decoding process, we ensure that the final molecule constructed from the tree is chemically valid and represents the behavior of our target GNN. Thank you for the opportunity to clarify this process. We have added an algorithm to describe this process in Appendix A.
>
> Q4. **Minor suggestions**
>
> Answer: We thank the reviewer for their detailed suggestions. We have addressed all inconsistent notations and corrected typographical errors in the revised paper.
>
> [1]. Jin, Wengong, Regina Barzilay, and Tommi Jaakkola. "Junction tree variational autoencoder for molecular graph generation." International conference on machine learning. PMLR, 2018.

---

> ### Author Response · Authors · 2024-11-25
> **Friendly Reminder**
>
> Dear Reviewer,
>
> Thank you again for your detailed review. In our rebuttal, we addressed the following key points:
>
> **Clarification of Notations:**  We updated and clarified symbols $L_{child}, L_{label}$ and provided detailed explanations for terms like $G$ (generated molecular graph) and $\mathcal{T}$ (predicted motif-node tree). These updates have been reflected in the revised manuscript.
>
> **Figure Improvements:** We clarified the depiction of Figure 3, confirming there is only one graph decoder and adjusted the figure to avoid misinterpretation.
>
> **Graph Construction from Subgraphs:** We provided a detailed explanation of our graph construction process from a tree structure, ensuring chemical validity and alignment with the target GNN. Additionally, we included an algorithm describing this process in the appendix A.
>
> **Minor Revisions:** We addressed all typographical errors and inconsistencies in notation as suggested.
>
> We would greatly appreciate your feedback on whether these clarifications and updates address your concerns. Thank you for your time and effort!

---

> ### Comment · Reviewer_9aZJ · 2024-11-25
> **Response to Authors**
>
> Thanks for the response and additional experiments.
>
> I think the response did not answer the question of the trivial solution, which states that the input graph is the output graph. At least from the evaluation metrics and the loss function, there is no regularization to avoid this solution.
>
> I noticed that the paper is updated. I suggest using another color to highlight the updated part.

---

> ### Author Response · Authors · 2024-11-25
> **Response to additional question**
>
> Thank you for your follow-up question and for pointing out the need for clarification regarding the trial solution where the input graph might trivially become the output graph. We appreciate the opportunity to address this concern further.
>
> 1. **Clarification on the generation process during inference:**
>
> During inference, our model uses the Junction Tree Variational Autoencoder (VAE) framework [1]. It starts with a random noise vector from the latent space, following a normal distribution. The tree decoder first generates a junction tree from this noise. Then, the graph decoder converts the tree into a graph. This process does not use any input graph, preventing a trivial solution. To provide further clarity, we have added Section 3.3.7 in our paper to elaborate on the explanation generation process.
>
> 2. **Evaluation of generated graphs using the novelty metric:**
>
> To ensure that our explainer does not simply regenerate graphs from the training set, we evaluated its outputs using the novelty metric, a widely used measure in the graph generation field [1, 2, 3]. This metric calculates the percentage of generated graphs that are entirely distinct from the training set, effectively assessing whether the generator is producing new examples instead of reproducing known ones.
>
> | Datasets     | Novelty of Label 0 | Novelty of Label 1 |
> |--------------|--------------------|--------------------|
> | Mutagenicity | 1.0                | 1.0                |
> | PTC_MR       | 1.0                | 1.0                |
> | PTC_MM       | 1.0                | 1.0                |
> | PTC_FM       | 1.0                | 1.0                |
> | AIDS         | 1.0                | 1.0                |
> | NCI-H23      | 1.0                | 1.0                |
>
> As shown in the table, our method achieves a 100\% novelty score across all six datasets. This result confirms that the generated explanations are not merely regenerating training examples but instead represent the learned properties of the target GNN.
>
> These findings further support that our framework generates meaningful and novel explanations aligned with the target GNN, rather than relying on trivial reproduction of training data. This discussion and result have been added in the Appendix H.
>
> 3. **Suggestion of highlighting updated part.**
>
> Thank you for your suggestion, we have updated our revised paper to highlight revised part in the paper with blue color.
>
> [1]. Jin, Wengong, Regina Barzilay, and Tommi Jaakkola. "Junction tree variational autoencoder for molecular graph generation." International conference on machine learning. PMLR, 2018.
>
> [2]. Jo, Jaehyeong, Seul Lee, and Sung Ju Hwang. "Score-based generative modeling of graphs via the system of stochastic differential equations." International conference on machine learning. PMLR, 2022.
>
> [3]. Jo, Jaehyeong, Dongki Kim, and Sung Ju Hwang. "Graph generation with diffusion mixture." arXiv preprint arXiv:2302.03596 (2023).

---

> > ### Comment · Reviewer_9aZJ · 2024-11-25
> > **Response to Authors**
> >
> > Thanks for the explanations. I don't have any further questions.

---

> ### Author Response · Authors · 2024-11-26
> **Thanks to Reviewer 9aZJ**
>
> We sincerely appreciate your thoughtful consideration of our responses and your kind reevaluation. Thank you for your valuable feedback and support, which have significantly contributed to refining our work.

---

### Official Review · Reviewer_jJYJ · 2024-11-02

**Soundness:** 4
**Presentation:** 3
**Contribution:** 4
**Rating:** 8
**Confidence:** 4

**Summary:**

This work proposes a novel method for generating model-level explanations by decomposing molecules into motifs and employing tree-constrained generators. To address issues of invalidity resulting from disregarding structural information, the method decomposes molecules into motif sets and uses attention-based motif identification to select key motifs for each class. A tree-constrained encoder-decoder generator and a specific loss function are introduced to ensure that the generated molecules conform to the class distribution, enhancing their validity. Experiments on six real-world datasets demonstrate that the proposed method outperforms the baselines in both effectiveness and efficiency. Qualitative results further highlight the importance of incorporating both node features and molecular structures in explanations.

**Strengths:**

1. This work clearly defines the problems faced by current model-level explanation methods and proposes a novel motif-based method to address them. This work conducts extensive experiments and analysis to support the claims, which are very solid and lay a good foundation for future studies in model-level molecular explanation.

2. By treating functional groups as building blocks, this method ensures that the explanations align more closely with scientifically meaningful interpretations. By considering both node features and molecular structures, the paper effectively addresses the limitations of atom-based methods.

3. The paper presents a novel approach that introduces an attention-based learning method to calculate the motif-class relationship.

4. The paper proposes using tree-constrained generators to produce more valid explanations. The carefully and explicitly designed tree decomposition and encoder-decoder structures ensure that the model generates more valid in-class molecules.

5. The evaluation metrics used to assess explanation performance in relation to molecules are better aligned with the chemical domain. The provided explanations are user-friendly and easy to understand.

6. The authors conduct experiments on six real-world datasets, and the results clearly demonstrate that the proposed method outperforms the baselines.

7. Comprehensive experimental settings are provided to ensure the quality and reproducibility of the results, and sufficient visualizations support the findings of the work.

**Weaknesses:**

1. There is a typo in lines 95 and 96 : there should be full stops at the end of the sentence.

2. In line 104, “adjacency” should be revised to “adjacency matrix.”

3. In line 168, “three methods” should be changed to “four methods.”

4. It would be beneficial to clearly state the limitations of current approaches and provide insights on how to address these limitations to better facilitate the development of this area within the community.

5. Additionally, summarizing and highlighting the main contributions and novelties of the proposed work would enhance clarity.

6. Furthermore, it would improve the presentation to arrange the notation used in the paper more effectively.

**Questions:**

1) Why is the bond feature defined as $N \times D_b$? The number of edges in a graph may not correspond to the number of nodes. How does this definition account for that discrepancy?

2) What distinguishes tree-constrained methods from non-constrained methods in molecular generation?

3) In the definition $T = (A_{\tau}, X_{\tau} )$, what does $A_{\tau}$  and $X_{\tau}$ represent? Additionally, what does $Z_{\tau}$ signify?

4) How is  $Z_{\tau}$ sampled from the graph encoder? What is the process involved?

5) What does  $f^a$ refer to in Section 3.3.5?

6) Why does XGNN experience out-of-memory (OOM) issues? Can the authors provide intuitive explanations based on experimental observations?

---

> ### Author Response · Authors · 2024-11-15
> **Response to Reviewer jJYJ**
>
> We sincerely thank the reviewer for the thorough and insightful review.
>
> Our responses to all questions are below:
>
> Q1. **Why is the bond feature defined as $N \times D_b$? The number of edges in a graph may not correspond to the number of nodes. How does this definition account for that discrepancy?**
>
> Answer: Thank you for pointing out this problem. We have revised bond feature as $|E| \times D_b$, where $|E|$ is the number of edges in the graph.
>
> Q2. **What distinguishes tree-constrained methods from non-constrained methods in molecular generation?**
>
> Answer: Thank you for raising this question. Upon reflection, we realize that the previous categorization of methods into tree-constrained and non-constrained may not have been entirely clear. To improve clarity, we have revised the categorization in the revised manuscript to focus on VAE-based methods and RL-based methods.
>
> VAE-Based Methods [1, 2] decompose and reconstruct molecules into a tree structure of motifs.
>
> RL-based methods [3] couple an RL policy network with fragment-based generation.
>
> Q3. **In the definition $\mathcal{T}$ = ($A_\mathcal{T}$, $X_\mathcal{T})$, what does $A_\mathcal{T}$ and $X_\mathcal{T}$ represent? Additionally, what does $z_{\mathcal{T}}$ signify?**
>
> Answer: Thank you for your question.
>
> $A_\mathcal{T}$ represents the adjacency matrix of the junction tree, which is derived from a molecular graph where each node corresponds to a motif.
>
> $X_\mathcal{T})$ represents the node feature matrix of the junction tree, where each node is a motif.
>
> $z_{\mathcal{T}}$ is a tree encoding.
>
> Q4. **How is $z_{\mathcal{T}}$ sampled from the graph encoder? What is the process involved?**
>
> Answer: Thank you for your question. I apologize for the unclear here. $z_{\mathcal{T}}$ sampled from the tree encoder similar to the process of $z_G$ sampled from the graph encoder. More specific, we use two different affine layers to compute the mean, represented as $ \mu_\mathcal{T}$, and the log variance, denoted as $ \log\sigma_\mathcal{T} $, from $ h_\mathcal{T} $. Then, we sample $ z_\mathcal{T} $ from a Gaussian distribution $ \mathcal{N} (\mu_\mathcal{T}, \sigma_\mathcal{T})$.
>
> Q5. **What does $f^a$ refer to in Section 3.3.5?**
>
> Answer: Thank you for your question. In Section 3.3.5, $f^a$ refers to a score function $f^a(G_i) = f(\phi(G_i))[r]$, where $r$ is the target label.
>
> Q6. **Why does XGNN experience out-of-memory (OOM) issues? Can the authors provide intuitive explanations based on experimental observations?**
>
> Answer: Thank you for raising this question. We are using the official implementation provided by the original paper to reproduce the results. Due to implementation inefficient, their official code cannot provide result on that dataset because they use a sparse adjacency matrix to store entire dataset. Based on our estimation, it requires 8T memory for NCIH123 dataset. Thus we can not provide comparison results with XGNN on NCIH23 dataset.
>
> Q7. **Typos**
>
> Answer: Thank you for pointing out all typos, we have fixed them in the revised paper.
>
> Q8. **Limitations of current approaches and highlight our contributions.**
>
> Answer: Thank you for these suggestions. We have highlighted our contributions and expanded the comparison between our method and existing literature in the introduction section.
>
> [1]. Jin, Wengong, Regina Barzilay, and Tommi Jaakkola. "Junction tree variational autoencoder for molecular graph generation." International conference on machine learning. PMLR, 2018.
>
> [2]. Jin, Wengong, Regina Barzilay, and Tommi Jaakkola. "Hierarchical generation of molecular graphs using structural motifs." International conference on machine learning. PMLR, 2020.
>
> [3]. Yang, Soojung, et al. "Hit and lead discovery with explorative rl and fragment-based molecule generation." Advances in Neural Information Processing Systems 34 (2021): 7924-7936.

---

> > ### Comment · Reviewer_jJYJ · 2024-11-22
> >
> > All my concerns have been addressed, and I recommend the paper for acceptance.

---

> > > ### Author Response · Authors · 2024-11-26
> > > **Thanks to Reviewer jJYJ**
> > >
> > > Thank you for thoughtfully reviewing our responses. We truly value your feedback and support.

---

### Official Review · Reviewer_KGJQ · 2024-11-03

**Soundness:** 1
**Presentation:** 2
**Contribution:** 1
**Rating:** 5
**Confidence:** 3

**Summary:**

This paper introduces MAGE, a motif-based approach to explain GNNs, specifically focused on molecular datasets. The approach addresses limitations in previous GNN explanation methods by employing molecular substructures—as foundational motifs in model explanations. MAGE utilizes a combination of motif extraction, attention-based motif learning, and a motif-based graph generation method to yield structurally valid and interpretable explanations at the model level. Experimental results on six molecular datasets show that MAGE achieves high validity and interpretability, outperforming baselines in providing explanations that are more representative.

**Strengths:**

- This paper is about a highly relevant topic in ML interpretability, particularly in the context of GNNs for molecular data. With the growing importance of explainable AI in sensitive domains, especially AI4Science domains like drug discovery and materials science, providing valid, interpretable explanations is well-aligned with current research trends and practical needs.

- The use of motifs as the basis for explanations addresses the limitations of atom-level generation methods.

- The paper is clear, especially regarding the MAGE’s workflow, from motif extraction to class-wise motif generation.

**Weaknesses:**

- The practical contribution of model-level GNN explanation, and its pros and cons compared to relevant works, say Q1 & Q2 below.

- No contribution to the computationally intensive and challenging motif extraction task.

- No human evaluation. Although the paper claims that the generated motifs and explanations are human-understandable, this claim is not supported by any human evaluation.

**Questions:**

1.	The paper emphasizes the practicality of model-level explanations over instance-level ones, yet in many real-world applications, users often seek to understand individual predictions. Could the authors elaborate on why a model-level approach would be more practical in such contexts and how it aligns with the needs of end-users who focus on instance-level interpretability?

2. How does this paper differentiate itself from MotifExplainer (Yu & Gao 2022), except for the "model-level" vs. "instance-level" difference that I am not convinced to believe is significant? MotifExplainer also utilizes motifs in GNN explanations. Can the authors clarify any unique aspects of MAGE, such as scope, or improvements in interpretability, validity, or computational efficiency?

3. MAGE’s approach begins by identifying all possible motifs in the dataset, which is a computationally intensive and challenging task, especially for large graphs or datasets due to the combinatorial explosion of possible motifs. However, MAGE does not contribute to addressing this fundamental challenge, as it relies on existing motif extraction methods without proposing any improvements to make motif identification more scalable. Addressing this limitation is more crucial to me, as the scalability of motif extraction represents a significant bottleneck for applying MAGE to larger datasets. Can the authors elaborate on the computational complexity of motif extraction and its impact on scalability to larger datasets?

5. Including a human evaluation would strengthen the claim by demonstrating that experts or users in the field find these explanations interpretable and meaningful. Any human evaluation to provide insights into the practical interpretability and further validate the effectiveness of MAGE in real-world applications?

6. Would MAGE be adaptable to non-molecular datasets, or are there constraints due to the specific nature of molecular motifs?

---

> ### Author Response · Authors · 2024-11-15
> **Response to Reviewer KGJQ (1)**
>
> We sincerely thank the reviewer for the thorough and insightful review.
>
> Our responses to all questions are below:
>
> Q1. **The paper emphasizes the practicality of model-level explanations over instance-level ones, yet in many real-world applications, users often seek to understand individual predictions. Could the authors elaborate on why a model-level approach would be more practical in such contexts and how it aligns with the needs of end-users who focus on instance-level interpretability?**
>
> Answer: Thank you for this thoughtful question. Model-level explanations offer practical benefits that can address broader concerns that may not be evident from instance-level insights alone.
> As noted by [1], "for example-level methods, we may need to explore the explanations for a large number of examples before we can trust the models. However, it is time-consuming and requires extensive expert efforts. For model-level methods, the explanations are more general and high-level, and hence need less human supervision."
> Similarly, [2] emphasize that "If the ultimate goal is to examine the model
> reliability, one will need to examine many instance-level explanations one by one to draw a rigorous
> conclusion about the model reliability, which is cumbersome and time-consuming. Conversely, the
> model-level explanation method can directly explain the high-level decision-making rule inside the
> blackbox (GNN) for a target prediction, which is less time-consuming and more informative regarding
> the trustworthiness of the GNNs. Besides, it has been shown that any instance-level explanation
> method would fail to provide a faithful explanation for a GNN that suffers from the bias attribution
> issue [3], while the model-level explanation methods can not only provide a faithful
> explanation for this case but also diagnose the bias attribution issue."
> [4] also mention that "Global Explainers, on the other hand,
> are aimed at capturing the behaviour of the model as a whole, abstracting individual noisy local
> explanations in favor of a single robust overview of the model".
>
> Ultimately, we see model-level and instance-level explanations as complementary. While our focus is on model-level explanations in this work.
>
> Q2. **How does this paper differentiate itself from MotifExplainer (Yu \& Gao 2022), except for the "model-level" vs. "instance-level" difference that I am not convinced to believe is significant? ...**
>
> Answer: Thank you for the insightful question. While both MAGE and MotifExplainer leverage motifs for GNN explanations, they differ significantly in both methodology and conceptual focus.
>
> 1. Model-Level vs. Instance-Level Distinction: Model-level and instance-level explanations are fundamentally different, especially in terms of their scope and applicability. Instance-level explanations, as used in MotifExplainer, focus on understanding model predictions for specific examples. This provides insight into individual decisions but may require extensive exploration across many examples to establish trust in the model’s general behavior. In contrast, MAGE’s model-level approach provides a holistic understanding of the model's decision-making patterns across classes. This model-level perspective enables users to quickly identify consistent motifs that influence model behavior, offering a more efficient and scalable solution for applications where interpretability across multiple instances is essential.
>
> 2. Methodological Differences in Using Motifs: Beyond the conceptual focus, MAGE and MotifExplainer also employ different methodologies. MAGE uses an attention-based method specifically for class-wise motif identification, enabling the identification of motifs that represent decision patterns across different classes. Additionally, MAGE’s use of a junction tree variational autoencoder (VAE) for generating explanations allows it to produce structurally coherent motifs that capture complex dependencies, leading to more realistic and domain-valid explanations.
>
> By combining these methodological and conceptual distinctions, MAGE offers strengths in interpretability and applicability for model-level insights.
>
> [1]. Yuan, Hao, et al. "Xgnn: Towards model-level explanations of graph neural networks." Proceedings of the 26th ACM SIGKDD international conference on knowledge discovery \& data mining. 2020.
>
> [2]. Wang, Xiaoqi, and Han-Wei Shen. "Gnninterpreter: A probabilistic generative model-level explanation for graph neural networks." arXiv preprint arXiv:2209.07924 (2022).
>
> [3]. Faber, Lukas, Amin K. Moghaddam, and Roger Wattenhofer. "When comparing to ground truth is wrong: On evaluating gnn explanation methods." Proceedings of the 27th ACM SIGKDD conference on knowledge discovery \& data mining. 2021.
>
> [4]. Azzolin, Steve, et al. "Global explainability of gnns via logic combination of learned concepts." arXiv preprint arXiv:2210.07147 (2022).

---

> ### Author Response · Authors · 2024-11-15
> **Response to Reviewer KGJQ (2)**
>
> Q3. **MAGE’s approach begins by identifying all possible motifs in the dataset, which is a computationally intensive and challenging task, especially for large graphs or datasets due to the combinatorial explosion of possible motifs. However, MAGE does not contribute to addressing this fundamental challenge, as it relies on existing motif extraction methods without proposing any improvements to make motif identification more scalable. Addressing this limitation is more crucial to me, as the scalability of motif extraction represents a significant bottleneck for applying MAGE to larger datasets. Can the authors elaborate on the computational complexity of motif extraction and its impact on scalability to larger datasets?**
>
> Answer: Thank you for your feedback regarding the scalability of motif extraction. We understand the importance of computational efficiency, especially for large datasets, and have taken measures to mitigate these concerns by selecting methods known for their speed and practicality. In particular, we use a bridge bond cut-off method [5], which is computationally efficient and straightforward to implement. This approach focuses on breaking bridge bonds to decompose molecules into meaningful substructures. The time complexity of used method is $O(V+E)$.
> Below, we provide a table illustrating the running times for decomposing molecules of different sizes using this method.
>
> |              | Average Number of Nodes | Motif Extraction Time (ms/graph) |
> |--------------|-------------------------|----------------------------------|
> | Mutagenicity | 30.23                   | 0.799                            |
> | PTC_MR       | 14.29                   | 0.428                            |
> | PTC_MM       | 13.97                   | 0.415                            |
> | PTC_FM       | 14.11                   | 0.444                            |
> | AIDS         | 15.69                   | 0.457                            |
> | NCI-H23      | 26.07                   | 0.877                            |
>
> This table demonstrates that the bridge bond cut-off method achieves decomposition times in the millisecond range. This approach allows us to process large datasets with minimal computational cost. We believe this choice effectively mitigates scalability concerns, ensuring that MAGE can be applied to real-world datasets without significant computational overhead. We have put this section into Appendix of our revised paper.
>
> Q4. **Including a human evaluation would strengthen the claim by demonstrating that experts or users in the field find these explanations interpretable and meaningful. Any human evaluation to provide insights into the practical interpretability and further validate the effectiveness of MAGE in real-world applications?**
>
> Answer: Thank you for this insightful suggestion. Following previous literature in the field[1, 2, 4], we mainly established quantitative and qualitative comparisons with existing methods to validate MAGE’s interpretability and effectiveness. However, we agree that a human evaluation would be a valuable addition to assess practical interpretability directly with domain experts. We consider this a promising plus to enhance the evaluation of this work further.
>
> Q5. **Would MAGE be adaptable to non-molecular datasets, or are there constraints due to the specific nature of molecular motifs?**
>
> Answer: Thank you for your question. Our method is adaptable to non-molecular datasets, provided there is a reliable motif extraction method specific to the dataset’s domain. MAGE relies on mutual motif extraction to identify recurring patterns that can be used for model-level explanations. In the molecular domain, well-established motif extraction methods [5, 6, 7] allow us to define and extract meaningful substructures, which form the basis of our explanations.
> For non-molecular datasets, a similar approach would be possible if there exists or can be developed a mutual motif extraction method tailored to the specific patterns within that dataset.
>
> [5]. Jin, Wengong, Regina Barzilay, and Tommi Jaakkola. "Hierarchical generation of molecular graphs using structural motifs." International conference on machine learning. PMLR, 2020.
>
> [6]. Lewell, Xiao Qing, et al. "Recap retrosynthetic combinatorial analysis procedure: a powerful new technique for identifying privileged molecular fragments with useful applications in combinatorial chemistry." Journal of chemical information and computer sciences 38.3 (1998): 511-522.
>
> [7]. Degen, Jorg, et al. "On the art of compiling and using'drug-like'chemical fragment spaces." ChemMedChem 3.10 (2008): 1503.

---

> > ### Comment · Reviewer_KGJQ · 2024-11-23
> >
> > Thank the authors for their rebuttal. My concerns are partially solved. However, I am still not fully convinced by the significance of the paper, especially the reliance on existing motif detectors, which may become a computational bottleneck and prohibit the framework from being extended to non-molecule data. I have updated my score to marginally below the acceptance.

---

> > > ### Author Response · Authors · 2024-11-24
> > > **Response to Additional Concerns**
> > >
> > > Answer: We appreciate your thoughtful feedback and are grateful for the opportunity to address your remaining concerns.
> > >
> > > **Clarification on Reliance on Motif Detectors**
> > >
> > > To clarify, motif extraction is part of the data preprocessing pipeline and is decoupled from the explainer training process. Once motifs are extracted, they are stored and can be reused across multiple training runs without incurring additional computational overhead. This separation ensures that motif detection does not become a bottleneck during training.
> > >
> > > **Extensibility Beyond Molecular Data**
> > >
> > > While our method is primarily designed for molecular datasets, it is not restricted to this domain. Existing graph-theoretic algorithms, such as gSpan [1] and MODA [2], can be employed to extract motifs for diverse types of graph data. For example, gSpan, a widely-used frequent subgraph mining algorithm, operates with time complexity of $O(kFS+rF)$, where $k$ is the maximum number of subgraph isomorphisms between a frequent subgraph and a graph in the dataset, $F$ is the number of frequent subgraphs, $S$ is the size of the dataset, and $r$ is the maximum number of duplicate codes of a frequent subgraph that grow from other minimum codes [1].
> > >
> > > As demonstrated in [1], gSpan processes datasets with approximately 340k graphs in just 10 minutes, showcasing its efficiency even for large-scale datasets. This efficiency makes it feasible to construct motif dictionaries for various graph types beyond molecular data, enabling the broader applicability of our framework.
> > >
> > > We believe this explanation further highlights the flexibility and scalability of our approach, mitigating concerns about computational bottlenecks.
> > >
> > > Once again, we sincerely appreciate your constructive comments and hope this response addresses your concerns fully.
> > >
> > > [1]. Yan, Xifeng, and Jiawei Han. "gspan: Graph-based substructure pattern mining." 2002 IEEE International Conference on Data Mining, 2002. Proceedings.. IEEE, 2002.
> > >
> > > [2] Omidi, Saeed, Falk Schreiber, and Ali Masoudi-Nejad. "MODA: an efficient algorithm for network motif discovery in biological networks." Genes \& genetic systems 84, no. 5 (2009): 385-395.

---

> > > ### Author Response · Authors · 2024-11-30
> > > **Friendly Reminder**
> > >
> > > Dear Reviewer:
> > >
> > > Thank you again for your thoughtful feedback and for giving us the opportunity to address your concerns in detail. We understand you may have a busy schedule, but we wanted to kindly check if our additional explanation has addressed your concerns.
> > >
> > > To summarize briefly, we clarified how motif extraction is part of the data preprocessing pipeline, ensuring it does not affect the explainer training process, and elaborated on the extensibility of the framework beyond molecular data using algorithms like gSpan and MODA. We hope these points adequately addressed your comments about computational bottlenecks and the broader applicability of our approach.
> > >
> > > If there are any remaining questions or concerns, we would greatly appreciate the chance to address them. We truly value your insights and feedback, which have been instrumental in refining our work.
> > >
> > > Thank you again for your time and consideration.

---

> > > ### Author Response · Authors · 2024-12-03
> > > **Follow-up with Reviewer KGJQ**
> > >
> > > Dear reviewer,
> > >
> > > As today is the last day to address any remaining concerns, we wanted to kindly follow up regarding my previous response to your thoughtful feedback. We understand you may have a busy schedule, but we wanted to check if my response addressed your concerns.
> > >
> > > If there are any further questions or points that require clarification, we would be more than happy to address them before the review period concludes. Thank you again for your time and valuable insights throughout this process.

---

### Official Review · Reviewer_MX2W · 2024-11-03

**Soundness:** 2
**Presentation:** 3
**Contribution:** 3
**Rating:** 6
**Confidence:** 4

**Summary:**

The paper proposes MAGE, a motif-based explanation method for GNNs in molecular tasks, which identifies significant motifs for each class using an attention-based learning approach. The method creates model-level explanations including critical molecular structures to the predictions. Experimental results show that MAGE provides valid, human-understandable explanations, outperforming SOTA baseline methods.

**Strengths:**

1) The paper is well-written, and its structure is easy to follow.
2) The central idea is clear and effectively delivered.
3) SOTA methods for model-level explainability are provided and compared in the experiments, enhancing the paper's credibility and impact.
4) The method generates valid molecules, which is not well-studied and essentially missing in the literature of GNN explainability. So I find this direction of research critical.

**Weaknesses:**

1) The code is not shared, which reduces the paper's reliability, especially given the extensive experiments presented.
2) The paper focuses only on model-level explainability baselines; however, local explainers (especially inductive ones) could potentially be adapted to the authors' chosen metric. Some local explainer baselines can be found in benchmarking study at ICLR 2024 [1].
3) The loss function comprises two main components, but the effectiveness of each part is not analyzed.
4) The examples from qualitative study is not well explained and confusing. It is hard to understand how are the examples selected.

Small:
- Line 244, typo: Figure 3.3.

[1] Kosan, M., Verma, S., Armgaan, B., Pahwa, K., Singh, A., Medya, S., & Ranu, S. GNNX-BENCH: Unravelling the Utility of Perturbation-based GNN Explainers through In-depth Benchmarking. In The Twelfth International Conference on Learning Representations.

**Questions:**

I do not have a major concern about the novelty of the paper. However, I have small but critical concerns about the paper's reproducibility and some experiment results. I'm willing to increase my score once my concerns are cleared.

1) Could you share the code to reproduce the experimental results?
2) Why does the baseline only include model-level explainers? Could local explainers be adapted to the same metrics? If not, can you explain the reasoning?
3) How does each component of the loss function impact the results?
4) How were example graphs, such as those in Table 3, selected? Is there any potential selection bias?

---

> ### Author Response · Authors · 2024-11-15
> **Response to Reviewer MX2W**
>
> We sincerely thank the reviewer for the thorough and insightful review.
>
> Our responses to all questions are below:
>
> Q1. **Could you share the code to reproduce the experimental results?**
>
> Answer: We thank the reviewer for pointing out this problem. We have uploaded our code to anonymous link: [https://anonymous.4open.science/r/MAGE-694E/](https://anonymous.4open.science/r/MAGE-694E/)
>
> Q2. **Why does the baseline only include model-level explainers? Could local explainers be adapted to the same metrics? If not, can you explain the reasoning?**
>
> Answer: We appreciate the reviewer’s suggestion regarding the adaptation of local explainers, particularly inductive ones, to our chosen metric. However, these instance level approaches fundamentally require an instance graph as input and the explainer is to extract most important subgraph as an explanation. This means they focus on explaining specific predictions for individual graphs rather than providing broader, generation-based model-level insights. For example, GEM which is an inductive instance-level explainer ``is to encourage a compact subgraph of the computation graph to have a large causal influence on the outcome of the target GNN"[1]. In addition, for instance-level explainer, there is no requirement for an explanation to be valid (e.g., a chemically valid molecular graph). To clarify this distinction, we have added a description of the differences between model-level explainers and inductive local explainers in the introduction section. (line 47-50)
>
> Q3. **How does each component of the loss function impact the results?**
>
> Answer: Thank you for pointing out this question. We conducted an ablation study on Mutagenicity dataset by evaluating performance when using each loss component independently. Below is the results.
>
> | Loss Function | Label 0             | Label 1             |
> |---------------|---------------------|---------------------|
> | $\mathcal{L_R}$           | 0.9827 $\pm$ 0.0580 | 0.9892 $\pm$ 0.0482 |
> | $\mathcal{L_P}$           | 0.9881 $\pm$ 0.0518 | 0.9805 $\pm$ 0.0589 |
> | $\mathcal{L_R} + \mathcal{L_P}$     | 0.9977 $\pm$ 0.0032 | 0.9941 $\pm$ 0.0240 |
>
> From the table, we observe that using only the reconstruction loss results in high accuracy, though it is slightly lower than configurations that include property alignment. This indicates that, while the model is proficient at preserving input structure, it may lack alignment with target-specific characteristics. The table also shows that using only the property loss results in slightly lower performance than the combined loss. This suggests that, without the reconstruction loss, the model may lose some structural accuracy, leading to a minor decrease in overall performance. We add this part as an ablation study in section 4.4.
>
> Q4. **How were example graphs, such as those in Table 3, selected? Is there any potential selection bias?**
>
> Answer: Thank you for your question. The example graphs for XGNN, GNNInterpreter, and Ours in Table 3 were selected randomly from the generated samples. To minimize any potential selection bias, we did not apply additional criteria beyond random sampling. Additionally, to further support our findings, we have included more examples in the appendix G, offering a broader representation of the generated outputs.
>
> Q5. **Typos**
>
> Answer: I have corrected typos in the revised paper.
>
> [1]. Lin, Wanyu, Hao Lan, and Baochun Li. "Generative causal explanations for graph neural networks." International Conference on Machine Learning. PMLR, 2021.

---

> ### Comment · Reviewer_MX2W · 2024-11-21
> **Thanks for the responses.  Additional questions and comments.**
>
> Thank you for your effort and for addressing my questions and concerns.
>
> Regarding Q2: Yes, you are correct. However, I believe they can still be compared to model-level explanations based on your metric definitions. For instance, you could take the average across different instances, etc. While local explainers are not trained to optimize this, it would be interesting to see how they perform in these scenarios. This comparison could also provide insight into the quality of model explainers in the context of model explanation metrics.
>
> Regarding Q4:
> - For Table 3, could you provide at least 3-5 random sample graphs? Providing only one example can be very misleading or results in a "lucky" graph.
> - For Table 8, could you clarify what each graph represents? It makes sense to have one graph per method since these are model-level explanations, but why are there multiple graphs for each class in Table 8? Providing more detailed explanations in the paper would be very helpful as well.

---

> > ### Author Response · Authors · 2024-11-21
> > **Response to additional questions and comments**
> >
> > We thank reviewer MX2W for additional discussion.
> >
> > Here are our answers to additional questions:
> >
> > Q1. **Yes, you are correct. However, I believe they can still be compared to model-level explanations based on your metric definitions. For instance, you could take the average across different instances, etc. While local explainers are not trained to optimize this, it would be interesting to see how they perform in these scenarios. This comparison could also provide insight into the quality of model explainers in the context of model explanation metrics.**
> >
> > Answer:
> > Thank you for your insightful comment. To address this, we performed an experiment on Mutagenicity dataset where we computed the Average Probability of two local explainers, GNNExplainer [1] and GEM [2], by averaging their probability outputs across various instances, as per your suggestion. The results for GNNExplainer were obtained using the PyTorch Geometric library, while GEM results were produced with its official implementation. The table below presents the findings.
> >
> > | Methods      | Label 0             | Label 1             |
> > |--------------|---------------------|---------------------|
> > | GNNExplainer | 0.8339 $\pm$ 0.1404 | 0.8250 $\pm$ 0.1402 |
> > | GEM          | 0.8739$\pm$ 0.1428  | 0.8887 $\pm$ 0.1306 |
> > | MAGE(Ours)   | 0.9977 $\pm$ 0.0032 | 0.9941 $\pm$ 0.0240 |
> >
> > From the results, it is evident that our MAGE approach outperforms the instance-level explainers significantly. This demonstrates the advantage of generating explanations from a global (model-level) perspective as opposed to relying on a local (instance-level) perspective. Model-level explanations generate insights leveraging global patterns and the overall model behavior, whereas instance-level explanations focus on generating explanations for individual data points or molecules. This distinction allows model-level explanations like MAGE to capture broader, more generalizable patterns within the dataset, which can provide a more holistic understanding of the model’s decision-making process. Conversely, instance-level methods such as GNNExplainer and GEM may provide more localized and specific insights, but often at the expense of lacking global context.
> >
> > We have added this section into appendix G in our revised paper.
> >
> > Q2. **For Table 3, could you provide at least 3-5 random sample graphs? Providing only one example can be very misleading or results in a "lucky" graph.**
> >
> > Answer: Thank you for your suggestion, we have added additional randomly selected sample graphs in Table 3 in our revised paper.
> >
> > Q3. **For Table 8, could you clarify what each graph represents? It makes sense to have one graph per method since these are model-level explanations, but why are there multiple graphs for each class in Table 8? Providing more detailed explanations in the paper would be very helpful as well.**
> >
> > Answer: Thank you for your question and for highlighting the need for clarification regarding Table 8. All generation-based model-level explainers can be viewed as generators, where varying the input noise results in different outputs. Consistent with the experimental setup in previous literature, we generated 1,000 explanation graphs for each class across six datasets when calculating the Average Probability metric.
> > To enhance clarity, we have added a detailed description of Table 8 in Appendix H, which provides further context and explanations.
> >
> > [1]. Ying, Zhitao, et al. "Gnnexplainer: Generating explanations for graph neural networks." Advances in neural information processing systems 32 (2019).
> >
> > [2]. Lin, Wanyu, Hao Lan, and Baochun Li. "Generative causal explanations for graph neural networks." International Conference on Machine Learning. PMLR, 2021.

---

> > > ### Comment · Reviewer_MX2W · 2024-11-22
> > > **Thanks for the clarification. Increasing my score.**
> > >
> > > I would like to thank the authors for their detailed explanations and efforts. With this version of the paper, I am now inclined to accept it and have decided to increase my score.

---

> > > > ### Author Response · Authors · 2024-11-22
> > > > **Thank to Reviewer MX2W**
> > > >
> > > > We sincerely appreciate your thoughtful review of our responses and your kind reconsideration in updating the evaluation. Thank you for your valuable feedback and support.

---

### Comment · Area_Chair_WChu · 2024-11-21
**Please engage with the authors in the discussion phase.**

Dear Reviewers,

The authors have posted a rebuttal to the concerns raised. I would request you to kindly go through their responses and discuss how/if this changes your opinion of the work.

best,

Area Chair

---

### Meta-Review · Area_Chair_WChu · 2024-12-08

**Metareview:**

The paper introduces a motif-based explanation method for Graph Neural Networks (GNNs) applied to molecular tasks. It employs an attention-based learning approach to pinpoint significant motifs for each class. This method provides model-level explanations that highlight crucial molecular structures influencing the predictions. Despite some concerns about its generalizability beyond molecular graphs, the reviewers unanimously recommend its acceptance.

**Additional Comments On Reviewer Discussion:**

Following the rebuttal, three reviewers supported the paper's acceptance, whereas Reviewer KGJQ expressed minor reservations about its generalizability beyond molecular graphs. However, during the discussion, the reviewer acknowledged that this limitation was not significant and agreed that the paper could be accepted.

---

### Decision · Program_Chairs · 2025-01-22

Accept (Poster)